

# Seasonally modulated nonlinear effects of PM₂.₅ on pediatric respiratory health: evidence from a time-series analysis in urban China

Weiqi Liu[1,*], Bingqing Liu[2,3,*], Weiling Liu[4], Liuhong Qu[5] and Cuiqing Qiu[6]

[1] Department of Clinical Laboratory, The Maternal and Children Health Care Hospital (Huzhong Hospital) of Huadu, Guangzhou, Guangdong, China
[2] Department of Pediatric, The Sixth Affiliated Hospital, Sun Yat-sen University, Guangzhou, China
[3] Biomedical Innovation Center, The Sixth Affiliated Hospital, Sun Yat-sen University, Guangzhou, China
[4] Department of Clinical Laboratory, Foshan Fosun Chancheng Hospital, Foshan, Guangdong, China
[5] Department of Pediatrics, Shenzhen New Frontier United Family Hospital, Shenzhen, Guangdong, China
[6] Medical Information Office, The Maternal and Children Health Care Hospital (Huzhong Hospital) of Huadu, Guangzhou, Guangdong, China
* These authors contributed equally to this work.

## ABSTRACT

**Background.** Although many studies have shown that fine particulate matter ($PM_{2.5}$) is associated with respiratory diseases (RDs) in children, fewer studies have examined this association in developing countries. We aimed to use the latest $PM_{2.5}$ standards to investigate the interaction between $PM_{2.5}$ and RDs among children in Guangzhou.

**Methods.** We included 18,291 pediatric inpatients aged 0–14 years with a primary diagnosis of RDs admitted to hospitals in Guangzhou, China, from January 1, 2018, to December 31, 2021. The association between $PM_{2.5}$ and RDs was analysed using a non-linear distributed lag model, and additional subgroup analyses were performed based on sex and season.

**Results.** The association of $PM_{2.5}$ with RDs showed a zigzag shape. Specifically, the cumulative effects of $PM_{2.5}$ at the 25th, 50th, and 75th percentiles, with a lag of 0–14 days, were associated with increases in the relative risk (RR) of RDs by 0.4% (95% confidence interval (CI) [1.000–1.007]), 3.4% (95% CI [1.004–1.065]), and 7.7% (95% CI [1.010–1.149]), respectively. Additionally, for each 10 μg/m³ increase in $PM_{2.5}$ concentration, the on-the-day lagged effect on lag day 4 and lag day 7 was associated with an elevated risk of RDs, with RR of 1.018 (95% CI [1.002–1.034]) and 1.016 (95% CI [1.000–1.032]), respectively. Additionally, $PM_{2.5}$ exposure significantly increased the risk of RDs in boys and elevated that risk in children during both summer and winter seasons.

**Conclusions.** This study reveals a significant effect of $PM_{2.5}$ exposure on RDs in children, with notably elevated risks during summer and winter seasons. These findings underscore the critical importance of implementing air quality improvement measures to safeguard children's health, particularly in developing countries.

Corresponding authors
Weiqi Liu, lwq_8103@163.com
Weiling Liu, 55768643@qq.com

## INTRODUCTION

Air pollutants are major health risks, driving global disease burdens, especially in low- and middle-income countries (*GBD 2015 Risk Factors Collaborators, 2016*). Particulate matter, especially that with an aerodynamic diameter equal to or less than 2.5 $\mu$m ($PM_{2.5}$), has been recognized as a major public health risk factor. Studies have shown that $PM_{2.5}$ has become the leading factor contributing to the global disease burden (*GBD 2021 Risk Factors Collaborators, 2024*). In China, for each 10 $\mu$g/$m^3$ increase in $PM_{2.5}$, the death risk of respiratory diseases (RDs) increases by 0.10% (*Liu et al., 2025*). Although the impact of $PM_{2.5}$ exposure on health has attracted attention around the world, nearly 99% of the world's population is still exposed to $PM_{2.5}$ levels higher than those of the 2021 update to the World Health Organization (WHO) Global Air Quality Guidelines (AQGs) (*Faridi et al., 2022*). It is estimated that the impacts of $PM_{2.5}$ on health will continue through 2030 (*Chi et al., 2024*).

Respiratory tract infection has become the disease with the highest morbidity in children due to the immature anatomical physiology and immune function of children's respiratory systems (*Ingham et al., 2019*). Pneumonia is the sixth leading cause of death in children under 5 years of age in China (*Faridi et al., 2022*; *He et al., 2017*). Owing to its rapid urbanization and industrial and economic development, China has paid insufficient attention to environmental pollution; this has resulted in serious urban environmental pollution, with adverse environmental, economic, and social effects of high $PM_{2.5}$ concentrations that have attracted the attention of local governments (*Wang et al., 2018a*). *Liu et al. (2022a)* conducted a study on $PM_{2.5}$ and RDs among children in Zhengzhou, China and found that $PM_{2.5}$ exposure was significantly related to RDs in children. Another researcher (*Liu et al., 2019*) studied the association between daily air pollution and hospitalization for RDs among children aged 0–17 years in Jinan, China. They found that $PM_{2.5}$ was significantly correlated with RD-related hospitalization among children. *Dong et al. (2021)* conducted a study on $PM_{2.5}$ and RDs among children aged 0–14 years in Lanzhou, China and found that $PM_{2.5}$ was significantly correlated with RDs.

While $PM_{2.5}$ levels in Guangzhou meet national standards, they remain higher than the $PM_{2.5}$ guideline limits recommended by the WHO in 2021. In Guangzhou, it has been reported that outdoor $PM_{2.5}$ concentrations were positively correlated with neonatal bacterial infection (*Liu et al., 2022b*). However, there are relatively few reports regarding the relationship between $PM_{2.5}$ and respiratory diseases. Therefore, this study aims to analyze the relationship between $PM_{2.5}$ and RDs in children in Guangzhou.

## MATERIALS & METHODS

### Study population

The daily hospitalization data for pediatric patients aged 0–14 years admitted from January 1, 2018 to December 31, 2021 were obtained from Maternal and Children Health Care Hospital of Huadu, which is a tertiary specialized hospital that mainly provides services for children and women, serving a population of more than 1.72 million. A tertiary hospital is the highest level of hospital classification in China. The diagnosis and classification

of patients in the hospital strictly follow the International Classification of Diseases 10th Revision (ICD-10) (*World Health Organization, 2016*). The patient's date of admission, sex, age, current address, and diagnosis of major diseases were obtained through the hospital's medical record management system. Children with RDs with ICD-10 diagnostic codes J00-J99 and who were local residents of the Huadu District of Guangzhou were included. This study was approved by the Ethics Committee of the Maternal and Children's Health Care Hospital of Huadu, Guangzhou (approval no. 2022-046). The committee dispensed with the need for informed consent due to the utilization of anonymized data.

## PM$_{2.5}$ exposure assessment

We collected the average daily real-time concentrations of PM$_{2.5}$ between January 1, 2018, and December 31, 2021, from National Environmental Automatic Monitoring Station No. 1351A (a pre-processed database, which is updated daily, is available at https://quotsoft.net/air). Station 1351A (113°12′52.56″E, 23°23′29.76″N) is situated within Huadu District, Guangzhou. It was established in compliance with China's Ministry of Ecology and Environment Technical Regulation on Ambient Air Quality Monitoring Site Selection (HJ 664-2013) and sited to represent regional background air quality across Huadu District. Consequently, we used its PM$_{2.5}$ concentrations as a proxy for population-level regional exposure in this study. Additionally, we confirmed that all study participants' residential addresses (obtained from hospital electronic medical records) were within Huadu District. The monitoring station is located approximately 4.2 km straight-line distance from the hospital, supporting the suitability of its data for assessing air pollution exposure in this study population.

## Covariates

Based on previous studies on air pollutants and RDs (*Tran et al., 2022*; *Wu et al., 2022*), potential confounding variables in this study were selected, including ozone (O$_3$) and carbon monoxide (CO), daily mean temperature (Temp), relative humidity (RH), average wind speed (WS), dew point temperature (DPT), day of the week (DOW), holidays, and season.

The average daily real-time concentrations of O$_3$ and CO were obtained from the same monitoring station as PM$_{2.5}$ (No. 1351A). Daily mean values of Temp, RH, WS, and DPT were obtained by selecting the meteorological monitoring point of Guangzhou Baiyun International Airport (113°17′56.400″E, 23°23′31.200″N). DOW is a dummy variable for day of the week; the variable holidays refers to Chinese legal holidays. Based on previous reports (*Rao et al., 2024*) and the geographical characteristics of Huadu District in Guangzhou, the seasonal divisions are as follows: spring spans from December to February, summer extends from March to May, autumn covers June to August, and winter occurs from September to November.

## Statistical analysis

In this study, descriptive statistics were used to analyze daily RDs hospitalizations, air pollutants, and meteorological factors; Spearman's rank correlation coefficients were computed. The distributed lag non-linear model (DLNM) was employed to analyze

the association between $PM_{2.5}$ exposure and RDs in children. Daily hospitalizations for RDs were designated as the dependent variable, while $PM_{2.5}$ concentration served as the independent variable. We adjusted for long-term time trends, $O_3$, CO, meteorological parameters, and other confounding factors. The annual time degree of freedom was determined to be 7 by selecting the minimum degree of freedom from the sum of the absolute values of partial autocorrelation of the residuals in the basic model. Based on previous research (*Jiang et al., 2021*; *Liu et al., 2022a*; *Qin et al., 2017*; *Wu et al., 2022*), the maximum lag time in this study was set to 14 days. The degrees of freedom for $O_3$, CO, and each meteorological parameter were set to 3 per year. Additionally, in accordance with the 2021 WHO Global Air Quality Guidelines, a $PM_{2.5}$ concentration of 15 $\mu g/m^3$ was used as the reference value in the model. The model was fitted using the quasiPoisson connection function and by controlling the time trend, DOW, holidays, and season. The basic model (model 1) is as follows:

$$\mathrm{Log}[E(Y_t)] = cb(X_i, lag) + ns(time, df) + DOW + holiday + season.$$

To evaluate the effects of $O_3$, CO, and meteorological parameters on $PM_{2.5}$ and RDs, $O_3$, CO, and meteorological parameters were introduced into the model as confounding factors, and we used the following modified model (model 2):

$$\mathrm{Log}[E(Y_t)] = cb(X_i, lag) + ns(time, df) + DOW + holiday + season + ns(O_3, df)$$
$$+ ns(CO, df) + ns(Temp, df) + ns(RH, df) + ns(WS, df) + ns(DPT, df)$$

where $t$ is the observation day, and $Y_t$ is the number of children with RDs on the observation day; $E(Y_t)$ is the expected number of children with RDs on day $t$; $X_i$ is $PM_{2.5}$, $cb()$ is the function of the two-dimensional cross-basis matrix generated by the DLNM model, *lag* indicates the lag days, *ns()* is the spline function, *df* is the degree of freedom, and *time* is used to control the long-term trend for the number of children newly diagnosed with RD each day. Furthermore, we conducted subgroup analyses by sex and season to explore potential variations in the effects of $PM_{2.5}$ on respiratory diseases across different quarters of the year.

Statistical analysis was performed using the "dlnm" package in R version 4.2.0 (The R Project for Statistical Computing), and the model was established to assess the association between $PM_{2.5}$ and RDs in children with a lag of 0–14 days. All statistical tests were two-sided, and $P$-values less than 0.05 were considered statistically significant.

## RESULTS

From 2018 to 2021, a total of 22,585 pediatric patients were hospitalized, among whom 18,291 were diagnosed with RDs, accounting for 80.99% of all hospitalized children. The daily respiratory disease hospitalizations ranged from 0 to 35, with the highest numbers observed in children aged $\leq 2$ years and males. Air pollution concentrations varied, with $PM_{2.5}$ ranging from 3.71 to 145.79 $\mu g/m^3$, $O_3$ from 3.88 to 175.67 $\mu g/m^3$, and CO from 0.21 to 26.00 $mg/m^3$. Meteorological factors showed a temperature range of 5.32–33.70 °C, relative humidity of 17.87%–97.25%, wind speed of 0.66–11.00 s/m, and dew point temperature from −9.77 °C to 26.65 °C (Table 1).

**Table 1  Summary of daily respiratory disease hospitalizations, air pollutants and meteorological factors in Guangzhou, from 2018 to 2021.**

| Variable | Min | Max | $P_{25}$ | $P_{50}$ | $P_{75}$ |
|---|---|---|---|---|---|
| Daily RDs number | 0.00 | 35.00 | 7.00 | 12.00 | 18.00 |
| Daily RDs number by Age (≤2 years) | 0.00 | 30.00 | 5.00 | 8.00 | 12.00 |
| Daily RDs number by Age (3–5 years) | 0.00 | 14.00 | 1.00 | 2.00 | 4.00 |
| Daily RDs number by Age (6–14 years) | 0.00 | 9.00 | 0.00 | 1.00 | 2.00 |
| Daily RDs number by Gender (male) | 0.00 | 26.00 | 4.00 | 7.00 | 11.00 |
| Daily RDs number by Gender (Female) | 0.00 | 19.00 | 2.00 | 4.00 | 7.00 |
| Daily RDs number by Year | | | | | |
| 2018 | 3.00 | 34.00 | 13.00 | 17.00 | 21.00 |
| 2019 | 5.00 | 35.00 | 13.00 | 18.00 | 23.00 |
| 2020 | 0.00 | 26.00 | 3.00 | 5.00 | 10.00 |
| 2021 | 0.00 | 21.00 | 6.00 | 8.00 | 10.00 |
| Daily RDs number by Season | | | | | |
| Spring (Dec–Feb) | 0.00 | 34.00 | 5.00 | 13.00 | 21.00 |
| Summer (Mar–May) | 0.00 | 35.00 | 5.00 | 10.00 | 18.00 |
| Autumn (June–Aug) | 0.00 | 30.00 | 8.00 | 12.00 | 14.50 |
| Winter (Sep–Nov) | 0.00 | 34.00 | 8.00 | 12.00 | 19.00 |
| Air pollution concentration | | | | | |
| $PM_{2.5}$ (μg/m$^3$) | 3.71 | 145.79 | 15.64 | 23.54 | 35.17 |
| $O_3$ (μg/m$^3$) | 3.88 | 175.67 | 32.76 | 49.83 | 69.38 |
| CO (mg/m$^3$) | 0.21 | 26.00 | 0.62 | 0.73 | 0.88 |
| Meteorological factors | | | | | |
| Temp (°C) | 5.32 | 33.70 | 19.49 | 24.68 | 28.49 |
| RH (%) | 17.87 | 97.25 | 61.35 | 70.11 | 77.27 |
| WS (s/m) | 0.66 | 11.00 | 1.96 | 2.50 | 3.38 |
| DPT (°C) | −9.77 | 26.65 | 13.10 | 19.15 | 23.93 |

**Notes.**

Abbreviations: RDs, respiratory diseases; $PM_{2.5}$, particulate matter with an aerodynamic diameter ≤ 2.5 μm; $O_3$, ozone; CO, carbon monoxide; Temp, temperature; RH, relative humidity; WS, wind speed; DPT, dew point temperature; Min, minimal value; Max, maximal value; $P_{25}$, 25th percentile; $P_{50}$, 50th percentile; $P_{75}$, 75th percentile.

Table 2 presents the Spearman correlation coefficients between air pollutants and meteorological factors in Guangzhou. $PM_{2.5}$ exhibited significant negative correlations with temperature ($\rho_s = -0.355$), wind speed ($\rho_s = -0.400$), and dew point temperature ($\rho_s = -0.443$) ($P < 0.05$), $O_3$ showed a significant positive correlation with temperature ($\rho_s = 0.239$; $P < 0.05$) and a weak positive correlation with CO ($\rho_s = -0.033$).

Adjusted for $O_3$, CO, and meteorological parameters, the curve between $PM_{2.5}$ and RDs formed a zigzag shape. With increased concentration of $PM_{2.5}$, the risk of RDs also increased; the risk of RDs decreased gradually with the extension of lag time, and the RD risk was the lowest at lag day 5 (Fig. 1).

Table 3 shows the cumulative effect of $PM_{2.5}$ exposure on RDs in percentiles of lag 0–14 days. After adjustment for $O_3$, CO, and meteorological parameters, the cumulative effects of $PM_{2.5}$ exposure at the 25th, 50th, and 75th percentiles (over lag days 0–14) were associated with increased risks of RDs, with relative risk (RR) of 1.004 (95% confidence interval
**Table 2    Spearman correlation coefficients between air pollutants and meteorological factors.**

| Variable | $PM_{2.5}$ | $O_3$ | CO | Temp | RH | WH | DPT |
|---|---|---|---|---|---|---|---|
| $PM_{2.5}$ | 1 | | | | | | |
| $O_3$ | 0.353* | 1 | | | | | |
| CO | 0.505* | −0.033 | 1 | | | | |
| Temp | −0.355* | 0.239* | −0.327* | 1 | | | |
| RH | −0.033 | −0.043 | −0.008 | −0.045 | 1 | | |
| WS | −0.400* | −0.121* | −0.236* | −0.067* | 0.039 | 1 | |
| DPT | −0.443* | −0.001 | −0.326* | 0.889* | −0.046 | −0.126* | 1 |

**Notes.**

Abbreviations: $PM_{2.5}$, particulate matter with an aerodynamic diameter $\leq$ 2.5 $\mu$ m; $O_3$, ozone; CO, carbon monoxide; Temp, temperature; RH, relative humidity; WS, wind speed; DPT, dew point temperature.

*$P < 0.05$.

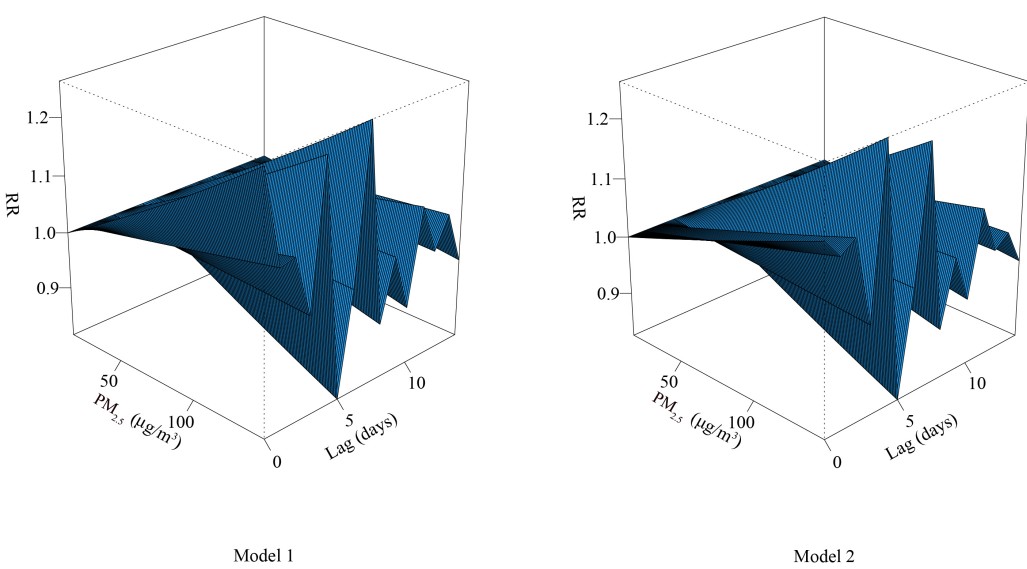

Model 1                                                    Model 2

**Figure 1    Three-dimensional diagram of exposure–response relationship between $PM_{2.5}$ and RDs.** Abbreviations: RR, relative risk. $PM_{2.5}$, particulate matter with an aerodynamic diameter $\leq$ 2.5 $\mu$ m.

(CI) [1.000–1.007]), 1.034 (95% CI [1.004–1.065]), and 1.077 (95% CI [1.010–1.149]), respectively.

Additionally, each 10-$\mu$g/m³ increase in $PM_{2.5}$ concentration was associated with a RR of 1.018 (95% CI [1.002–1.034]) for RDs at lag day 4 and 1.016 (95% CI [1.000–1.032]) at lag day 7, indicating a significant increase in the risk of RDs (Fig. 2).

As shown in Fig. 3, a per 10-$\mu$g/m³ increase in $PM_{2.5}$ concentration at lag 0–14 days was associated with a cumulative RR for RDs of 1.074 (95% CI [1.009–1.143]), indicating a statistically significant increased risk. In males, the cumulative RR was 1.046 (95% CI [1.005–1.089]), also reflecting a significant increase in risk. By contrast, the association was not statistically significant in females.

After adjusting for confounding factors, each 10-$\mu$g/m³ increase in the concentration of $PM_{2.5}$ with a lag of 0–14 days was associated with an 85.7% higher risk of RDs in summer

**Table 3** The cumulative effect of PM$_{2.5}$ exposure on RDs in percentiles of lag 0–14 days.

| Variable | Model 1 | | | Model 2 | | |
|---|---|---|---|---|---|---|
| | RR | 95% CI | *P*-value | RR | 95% CI | *P*-value |
| $P_{25}$ | 1.004 | 1.001–1.007 | <0.05 | 1.004 | 1.000–1.007 | <0.05 |
| $P_{50}$ | 1.034 | 1.007–1.062 | <0.05 | 1.034 | 1.004–1.065 | <0.05 |
| $P_{75}$ | 1.077 | 1.016–1.143 | <0.05 | 1.077 | 1.010–1.149 | <0.05 |

**Notes.**

Abbreviations: RR, Relative risk; 95% CI, 95% confidence interval; $P_{25:}$, 25th percentile; $P_{50:}$, 50th percentile; $P_{75:}$, 75th percentile.

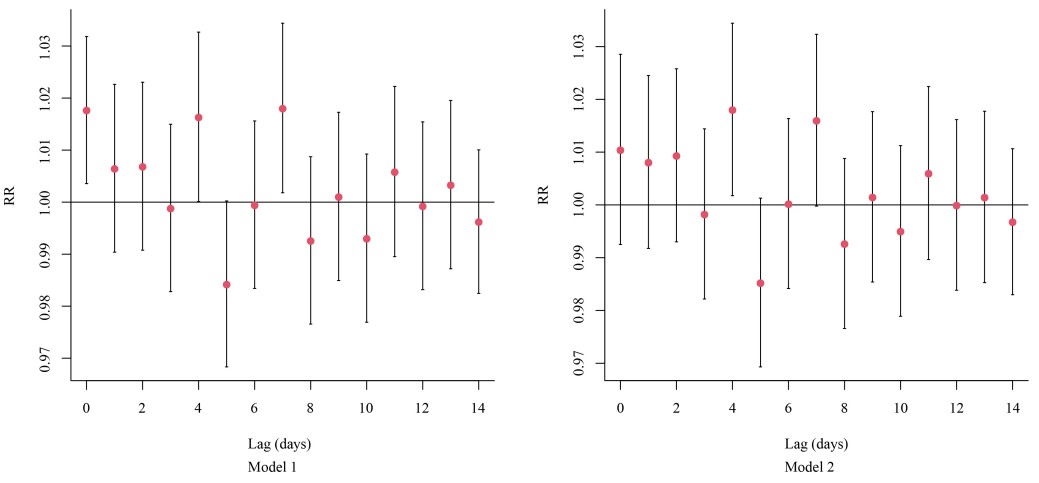

**Figure 2** The on-the-day lagged effects per 10-μg/m$^3$ increase in the concentration of PM$_{2.5}$ 2.5 on RDs. Abbreviations: RR, relative risk.

(RR = 1.857; 95% CI [1.077–3.203]) and a 5.2% higher risk in winter (RR = 1.052; 95% CI [1.013–1.092]). In boys, each 10-μg/m$^3$ increase in PM$_{2.5}$ exposure was associated with a 113% higher risk of RDs during summer (RR = 2.130; 95% CI [1.064–4.264]) and a 5.7% higher risk during winter (RR = 1.057; 95% CI [1.008–1.107]) (Table 4).

## DISCUSSION

In this study, we conducted a time-series analysis of the relationship between PM$_{2.5}$ and RDs among children in a children's hospital in Guangzhou. Our findings indicate that PM$_{2.5}$ exposure substantially increases the risk of RDs, and the lagged effect reaches its peak on days 4 and 7. The increase in risk is more pronounced among male children. Additionally, this study found that the risk of RDs in children was higher in summer and winter. These findings highlight the necessity for targeted air quality management and public health interventions to reduce the risk of RDs.

Previous studies have demonstrated that exposure to PM$_{2.5}$ increases the susceptibility of the respiratory system to various pathogens (*Yang, Li & Tang, 2020*). Research indicates that, among environmental health risks, air pollution—particularly PM$_{2.5}$—presents the

| Variable | RR(95%CI) |
|---|---|
| **All** | |
| Model 1 | 1.073 (1.013-1.135) |
| Model 2 | 1.074 (1.009-1.143) |
| **Male** | |
| Model 1 | 1.035 (0.998-1.074) |
| Model 2 | 1.046 (1.005-1.089) |
| **Female** | |
| Model 1 | 1.042 (0.995-1.092) |
| Model 2 | 1.026 (0.976-1.079) |

**Figure 3** Forest plot of the cumulative effect of each 10-$\mu$g/m$^3$ increase in the PM$_{2.5}$ concentration on RDs. Abbreviations: RR, relative risk. 95% CI, 95% confidence interval.

**Table 4 Impact of each 10-$\mu$g/m$^3$ increase in PM$_{2.5}$ on RD in different seasons, from 2018 to 2021.**

| Variable | RR (95% CI) | | |
|---|---|---|---|
| | **All** | **Male** | **Female** |
| **Spring** | | | |
| Model 1 | 1.129 (0.926–1.377) | 1.127 (0.864–1.471) | 1.145 (0.829–1.581) |
| Model 2 | 0.989 (0.791–1.236) | 0.964 (0.713–1.305) | 1.037 (0.720–1.492) |
| **Summer** | | | |
| Model 1 | 1.821 (1.136–2.919) | 1.998 (1.094–3.649) | 1.591 (0.730–3.466) |
| Model 2 | 1.857 (1.077–3.203) | 2.130 (1.064–4.264) | 1.578 (0.650–3.834) |
| **Autum** | | | |
| Model 1 | 1.130 (0.865–1.476) | 1.134 (0.801–1.606) | 1.168 (0.774–1.764) |
| Model 2 | 1.205 (0.886–1.638) | 1.345 (0.905–1.997) | 1.114 (0.686–1.809) |
| **Winter** | | | |
| Model 1 | 1.050 (1.014–1.087) | 1.042 (0.998–1.089) | 1.061 (1.004–1.120) |
| Model 2 | 1.052 (1.013–1.092) | 1.057 (1.008–1.107) | 1.044 (0.985–1.106) |

Notes.
Abbreviations: RR, Relative risk; 95% CI, 95% confidence interval.

most significant threat concerning respiratory issues in infants and children (*Jakubiak-Lasocka, Lasocki & Badyda, 2015*). *He et al. (2022)* found that PM$_{2.5}$ not only affects emergency room visits for all respiratory diseases on the day of exposure but also exhibits

a cumulative effect. In a study involving 332,337 children (aged 0–13 years) in Lanzhou, the results demonstrated a significant correlation between increased $PM_{2.5}$ levels and a rise in outpatient visits for overall respiratory diseases among children (*Ma et al., 2020*). A key strength of this study lies in its specific focus on children within the unique context of Guangzhou, a densely populated subtropical metropolis with distinct air pollution patterns and a large migrant population. Our findings align well with previous studies, indicating that the risk of respiratory diseases (RDs) increases with rising concentrations of $PM_{2.5}$. This consistency can be attributed to several factors, primarily the anatomical and physiological immaturity of children's respiratory systems, rendering them more susceptible to external environmental factors, especially particulate matter. Additionally, children's immune systems are still in the developmental stage, conferring a relatively weaker capability to fend off external pathogens and pollutants. Consequently, in environments with high $PM_{2.5}$ levels, children are particularly vulnerable, leading to an augmented risk of developing respiratory diseases. This comprehensive understanding underscores the necessity for targeted public health strategies aimed at mitigating the adverse effects of air pollution on pediatric respiratory health.

A study conducted in Yichang City, China, demonstrated that for every interquartile range increase in $PM_{2.5}$ levels with a lag of 0 days, there was increase in pediatric respiratory outpatient visits (*Liu et al., 2017*). A study involving 39,766 children in Taiwan found that the RR of acute upper respiratory infections due to $PM_{2.5}$ exposure peaked at a lag of 4 to 5 days (*Lin et al., 2013*). Research conducted in Shenzhen, China, indicated that the risk of hospitalization for respiratory conditions significantly increased due to $PM_{2.5}$ exposure after a lag of five days (*Liang et al., 2021*). However, a study in Poland found no significant increase in the risk of upper respiratory infections from short-term $PM_{2.5}$ exposure within a lag of 0 to 6 days (*Ratajczak et al., 2021*). This study also showed that there is a lag effect between $PM_{2.5}$ and RDs in children. $PM_{2.5}$ exposure had the lowest effect on RDs in children with a time lag of 5 days, which is inconsistent with earlier studies (*Liang et al., 2021*; *Slama et al., 2019*). Inconsistencies in findings may be attributed to several factors. Geographical and climatic differences, such as variations in temperature, humidity, and wind speed, influence the dispersion and retention of $PM_{2.5}$, thereby affecting its health impacts on respiratory diseases (*Garsa et al., 2023*; *Li et al., 2019*; *Li, Liu & Zhao, 2022*). Furthermore, differences in health status, lifestyle, and access to healthcare across populations may modify the association between $PM_{2.5}$ exposure and respiratory diseases. For example, Guangzhou differs demographically from cities such as Yichang, Taiwan, and Poland in its substantial population of migrant children, who often belong to lower socioeconomic groups and have severely limited access to healthcare services. This subgroup may be more susceptible to the adverse effects of air pollution due to underlying health inequities. Moreover, delayed medical care may alter the timing and severity of disease onset, thereby amplifying the observed health impacts of $PM_{2.5}$ exposure in this population. Collectively, these elements contribute to the variability observed in research outcomes.

A study in Changchun, China showed that when the concentration of $PM_{2.5}$ increased by $10\ \mu g/m^3$, the risk of hospitalization for RDs increased by 0.31% (*Jia et al., 2022*). A study of the effects of air pollutants and meteorological conditions on hospitalization for RDs in

Shenzhen, China found that when the concentration of $PM_{2.5}$ increased by 10%, the risk of hospitalization for RDs increased by 0.23% (*Liang et al., 2021*). The results of a time-series study conducted in Yichang City, China showed that with each interquartile range increase in $PM_{2.5}$ concentrations, there was a 1.91% increase in pediatric respiratory outpatient visits (*Liu et al., 2017*). The results of a meta-analysis indicated that for each 10 µg/m$^3$ increase in $PM_{2.5}$, the risk of chronic obstructive pulmonary disease (COPD)-related emergency department (ED) visits and outpatient clinic visits increased by 2.5% (95% CI [1.6%–3.4%]) (*DeVries, Kriebel & Sama, 2017*). The results of the present study were significantly higher than those of previous studies, with the risk of RDs increasing by 7.4% per 10-µg/m$^3$ increase in the concentration of $PM_{2.5}$, which is slightly different from other studies. The differences in results may be attributed to several factors. Firstly, the study populations varied: studies conducted in Changchun and Shenzhen focused on adults and children, whereas our study targeted children exclusively. Secondly, there were differences in the research methodologies employed. Our study incorporated a broader range of meteorological parameters, whereas previous studies often considered only one or two factors. The combined effects of these meteorological factors can exacerbate $PM_{2.5}$ pollution, thereby increasing the risk of respiratory and other related diseases. These differences in study populations and methodologies may have led to variations in results.

An increasing number of studies have shown a link between air pollution and RDs in both sexes, but the results are not completely consistent. A study conducted by *Hu et al. (2024)* in China showed that males are more vulnerable to air pollutants. *Wang et al. (2018b)*. studied the effects of air pollutants on RDs in Shandong Province, China. The results showed that air pollutants had significant effects on both men and women. Our study results showed that male children exposed to $PM_{2.5}$ were more prone to RDs than female children. This may be owing to differences between male and female airways (*Becklake & Kauffmann, 1999*; *Dong et al., 2011*; *Polgar & Weng, 1979*), male children have more immature lungs and relatively narrow airways in childhood, making them more prone to RDs than female children.

Research has shown that $PM_{2.5}$ is closely linked to the exacerbation of respiratory diseases in winter (*Romaszko-Wojtowicz et al., 2025*). The findings from a study encompassing 372 cities across 19 countries and regions indicate that the impact of $PM_{2.5}$ on mortality due to respiratory diseases is significantly more pronounced during the cold season (*Liu et al., 2023*). However, this study reveals an increased risk of respiratory diseases among children during both the summer and winter seasons. The observed inconsistencies may be attributed to the study area being located in a subtropical zone, where hot and humid summers can impede the dispersion of $PM_{2.5}$. Elevated temperatures also enhance photochemical reactions, contributing to the formation of ozone and other secondary pollutants, thereby further deteriorating air quality. Furthermore, individuals across different regions adopt varying protective measures during different seasons, which can influence the study outcomes.

This study has several strengths. First, by focusing on the pediatric population in the subtropical megacity of Guangzhou, it provides critical evidence for developing strategies to mitigate the health impacts of $PM_{2.5}$ exposure on children with respiratory diseases in

similar metropolitan settings. Second, time-series analysis effectively captures significant lagged effects of $PM_{2.5}$ exposure and reveals distinct seasonal patterns in associated health risks. However, this study has several limitations. First, the sample was derived exclusively from clinical records of a single children's hospital in Guangzhou, restricting the population to pediatric patients without comparative data from other age groups; this limited representativeness may constrain the generalizability of findings. Second, although adjustments were made for confounders including pollutants and meteorological factors, residual confounding may persist due to the model's inability to fully control for potential confounders such as indoor air pollution exposure, personal protective behaviors, and socioeconomic status. Furthermore, modeling season as a categorical variable in this study potentially oversimplifies its continuous and dynamic nature, failing to capture intra-seasonal fluctuations and complex interactions with other environmental factors, thereby compromising the accurate assessment of real-world health risks. Finally, individual $PM_{2.5}$ exposure was estimated using fixed-site monitoring concentrations, which neglects variations in personal activity patterns, microenvironmental differences, and spatial heterogeneity. This approach may introduce non-differential misclassification in exposure assessment, potentially biasing effect estimates. Despite the aforementioned limitations, these findings highlight the urgency of implementing effective measures to reduce air pollution levels. It is recommended to strengthen industrial emission controls, accelerate the transition to clean energy, and optimize urban public transportation systems to mitigate pollutant emissions at the source. Concurrently, widespread implementation of vaccination strategies against respiratory infections should be promoted to enhance immune protection in children (*Cattaneo, 1994*; *Ding et al., 2025*). Coordinated efforts in environmental and public health interventions are essential to mitigate the long-term adverse effects of air pollution on children's respiratory health.

## CONCLUSIONS

In summary, this study demonstrates a significant association between $PM_{2.5}$ exposure and an increased risk of respiratory conditions in children, particularly during the summer and winter seasons. These findings emphasize the urgent need for targeted public health interventions to mitigate the adverse effects of air pollution on children's respiratory health.

### Funding
The authors received no funding for this work.

### Competing Interests
The authors declare there are no competing interests.

## Author Contributions

- Weiqi Liu conceived and designed the experiments, performed the experiments, analyzed the data, prepared figures and/or tables, authored or reviewed drafts of the article, and approved the final draft.
- Bingqing Liu performed the experiments, authored or reviewed drafts of the article, and approved the final draft.
- Weiling Liu conceived and designed the experiments, performed the experiments, authored or reviewed drafts of the article, and approved the final draft.
- Liuhong Qu analyzed the data, prepared figures and/or tables, and approved the final draft.
- Cuiqing Qiu performed the experiments, prepared figures and/or tables, and approved the final draft.

## Human Ethics

The following information was supplied relating to ethical approvals (i.e., approving body and any reference numbers):

The Ethics Committee of the Maternal and Children's Health Care Hospital of Huadu approval to carry out the study (approval no. 2022-046).

## Data Availability

The raw data are available in the Supplementary File.

## Supplemental Information

Supplemental information for this article can be found online at http://dx.doi.org/10.7717/peerj.20187#supplemental-information.

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
