# Peer review of "Seasonally modulated nonlinear effects of PM2.5 on pediatric respiratory health: evidence from a time-series analysis in urban China"

_PeerJ, doi:10.7717/peerj.20187_

## Round 0.1 · original submission · Minor Revisions

**Language Note:** When you prepare your next revision, please either (i) have a colleague who is proficient in English and familiar with the subject matter review your manuscript, or (ii) contact a professional editing service to review your manuscript. PeerJ can provide language editing services - you can contact us at [email protected] for pricing (be sure to provide your manuscript number and title). – PeerJ Staff

Reviewer 1 ·

Basic reporting

What are the effects of meteorological factors on diseases?
 Mention the vaccination and necessary policy on air pollution to control the diseases in children.

Experimental design

-

Validity of the findings

-

Reviewer 2 ·

Basic reporting

1. The English language in the abstract should be improved for clarity and readability. For example, the sentence “Specifically, the cumulative effects of PM2.5 at the 25th, 50th, and 75th percentiles with a lag of 0-14 days, the relative risk of RD increased…” and the following sentence “Each 10 g/m³ increase in PM2.5 concentration, the on-the-day lagged effect value of RDs on lag day 4 and lag day 7 were 1.018 (95% CI: 1.002–1.034) and 1.016 (95% CI: 1.000–1.032).” should be revised for clarity.

2. The discussion in lines 202-209 would benefit from the inclusion of relevant references to support the statements. Providing citations to previous studies with similar findings or methodological considerations would help strengthen the scientific credibility of the discussion.

Experimental design

1. Please add a description of the correlation analysis (line 148) in the Methods section. This would help clarify how potential multicollinearity among variables was assessed and, if applicable, how it influenced subsequent analyses.

2. For the model in line 124, it is worth considering whether using a categorical variable “season” is an appropriate method for adjusting for seasonal variation. This approach may oversimplify the complex and continuous nature of seasonal influences. I suggest conducting sensitivity analyses using alternative methods to account for seasonality, for example, by adjusting temperature-related variables (Yu et al., PLoS Med, 2022).

3. Given that previous studies have used varying lag structures to assess the effects of PM2.5 on respiratory diseases, I suggest the authors conduct sensitivity analyses using alternative lag days. This would help evaluate the robustness of the observed associations to different lag specifications.

Validity of the findings

1. Please ensure that the results are described accurately. In line 160, the relative risk values for lag day 4 and lag day 7 are reported as 1.018 and 1.016, respectively. However, these values do not appear to correspond to the data presented in Figure 2.

Reviewer 3 ·

Basic reporting

Proper English and grammar were used throughout. References were properly added and up-to-date. Tables and figures were included and cited in-text properly.

One typo found: Line 174-175: should be “more pronounced among male children and shows seasonal variation.”

Experimental design

- In the article, it is mentioned that the air monitor chosen is near the location of the hospital as a measure of exposure. However, it is not mentioned whether the monitor and hospital are also near the location of patients' residences. Are we to assume that patients lived near the hospital so their exposure would be properly captured by that air monitor? Could it be possible that patients came from different regions and had different exposure levels? If possible, patients' geographic residence and proximity to the air monitor should also be considered as potential confounders.

- There is no mention in the article of what is considered a "pediatric population." This can change from hospital to hospital. Some studies report 0-18 pediatric, while others may consider 0-19 or 0-21 as pediatric. That should be mentioned.

- Also, there is no mention of demographic characteristics of the sample. A table describing how many patients fall into different age groups or the mean age group of the population would be helpful to understand the sample.

- Children's health depends heavily on the age group they fall into. Many studies have reported that within a pediatric population, children ages 0-5 have worse impacts compared to older children after exposure to environmental hazards. This analysis would benefit from a sensitivity analysis that stratifies by age group to understand age group differences.

- Seems like confounding variables were only considered in terms of the exposure, which is good, but should also be considered for the outcome. Factors like age group, previous history of asthma/other respiratory conditions in children, should be considered, as asthmatic children are a vulnerable population.

- The author discusses population disparities in the region, but doesn’t go into depth about the specific differences this region may have in terms of population disparities compared to other regions, which may explain the inconsistencies. In the discussion section, I would recommend a bit more information about the demographics of the region and how that makes them more vulnerable to the effects of air pollution, and less about previous studies.

Validity of the findings

-

Additional comments

- This paper would benefit from highlighting a few strengths of the study and the implications of the results from the area. In the discussion, it could elaborate on how the results can be used for future studies.

---

## Round 0.2 · accepted · Accept

This revised version is suitable for publication.

Reviewer 2 ·

Basic reporting

The revised manuscript is clear and professional.

Experimental design

The experimental design meets the required standards.

Validity of the findings

The analyses well support the validity of the findings.

Reviewer 3 ·

Basic reporting

-

Experimental design

-

Validity of the findings

-

Additional comments

All my previous comments have been adequately addressed, and I believe the manuscript is ready for submission.